# Molecular assembly of the period-cryptochrome circadian transcriptional repressor complex

**Shannon N Nangle[1†], Clark Rosensweig[2†], Nobuya Koike[3], Hajime Tei[4], Joseph S Takahashi[2,5]\*, Carla B Green[2]\*, Ning Zheng[1,6]\***

[1]Department of Pharmacology, University of Washington, Seattle, United States; [2]Department of Neuroscience, University of Texas Southwestern Medical Center, Dallas, United States; [3]Department of Physiology and Systems Bioscience, Kyoto Prefectural University of Medicine, Kyoto, Japan; [4]Graduate School of Natural Science and Technology, Kanazawa University, Ishikawa, Japan; [5]Howard Hughes Medical Institute, University of Texas Southwestern Medical Center, Dallas, United States; [6]Howard Hughes Medical Institute, University of Washington, Seattle, United States

**\*For correspondence:** joseph. takahashi@utsouthwestern.edu (JST); carla.green@ utsouthwestern.edu (CBG); nzheng@u.washington.edu (NZ)

[†]These authors contributed equally to this work

**Competing interests:** The authors declare that no competing interests exist.

**Reviewing editor**: Louis Ptáček, University of California, San Francisco, United States

**Abstract** The mammalian circadian clock is driven by a transcriptional–translational feedback loop, which produces robust 24-hr rhythms. Proper oscillation of the clock depends on the complex formation and periodic turnover of the Period and Cryptochrome proteins, which together inhibit their own transcriptional activator complex, CLOCK-BMAL1. We determined the crystal structure of the CRY-binding domain (CBD) of PER2 in complex with CRY2 at 2.8 Å resolution. PER2-CBD adopts a highly extended conformation, embracing CRY2 with a sinuous binding mode. Its N-terminal end tucks into CRY adjacent to a large pocket critical for CLOCK-BMAL1 binding, while its C-terminal half flanks the CRY2 C-terminal helix and sterically hinders the recognition of CRY2 by the FBXL3 ubiquitin ligase. Unexpectedly, a strictly conserved intermolecular zinc finger, whose integrity is important for clock rhythmicity, further stabilizes the complex. Our structure-guided analyses show that these interspersed CRY-interacting regions represent multiple functional modules of PERs at the CRY-binding interface.

## Introduction

Life on Earth evolved a self-sustaining molecular timing system that synchronizes cellular activities with the solar day. This endogenous clockwork prepares an organism for periodic environmental fluctuations and coordinates numerous physiological and behavioral processes (*Reppert and Weaver, 2002*). At the molecular level, the mammalian circadian clock operates through an auto-regulatory transcription–translation feedback loop composed of four core components—the transcriptional activator proteins, CLOCK and BMAL1, and the transcriptional repressors, Periods (PERs) and Cryptochromes (CRYs). The heterodimeric CLOCK and BMAL1 complex acts as the positive arm of the loop by recognizing E-box elements and promoting the expression of clock-controlled genes, including *Per1*, *Per2*, *Cry1*, and *Cry2*. The PER and CRY proteins function as the negative arm of the loop by blocking the activity of CLOCK-BMAL1 and inhibiting the transcription of their own and all other clock-controlled genes. The cyclic accumulation, localization, and degradation of the PER and CRY proteins are necessary to manifest a 24-hr rhythm (*Lowrey and Takahashi, 2011*).

Earlier studies suggested that CRYs are the predominant inhibitors of CLOCK-BMAL1 (*Griffin et al., 1999*; *Kume et al., 1999*). Independent of PERs, overexpressed CRY1 and CRY2 can each potently inhibit

**eLife digest** Since the very simplest organisms emerged on earth, the rhythms of life have been synchronized with the rising and setting of the sun. Even the most basic life forms have internal clocks that help them to maintain daily routines and adapt to shifting seasons. In animals, these internal clocks regulate processes such as the release of hormones that wake an animal up and the expression of genes necessary to carry out the activities of daily life. Later on, the clocks then trigger the release of hormones that cause drowsiness and the expression of the genes that are active during rest.

In mammals, these internal circadian rhythms are maintained by a feedback loop governed by four key proteins. Two of these proteins—CLOCK and BMAL1—work together to begin a process called transcription, whereby sections of DNA are used as a template to copy the information needed to make a protein. The two activating proteins CLOCK and BMAL1 recognize the sections of DNA where the genes that are controlled by the circadian clock are located and selectively turn on the expression of those genes.

Expression of the two other key circadian proteins—Period and Cryptochrome—is switched on by CLOCK and BMAL1. As Period and Cryptochrome proteins accumulate, they begin to inhibit the activity of CLOCK and BMAL1, helping to reduce the rate at which the circadian genes are transcribed as the day progresses.

Nangle et al. provide new insights into how the Period and Cryptochrome proteins interact with each other, using X-ray crystallography to reveal the molecular level details of the bond between the two proteins. Period stretches out as it 'embraces' Cryptochrome. One end of the Period protein then tucks into part of the Cryptochrome structure that is next to a large pocket. This pocket is where the Cryptochrome protein binds to CLOCK and BMAL1, suggesting that Period can influence whether this binding occurs.

The other end of the Period protein covers one end of the Cryptochrome protein. By doing so, enzymes cannot bind there, and so cannot break down Cryptochrome. Nangle et al. also discovered that a finger-like projection that includes a zinc ion acts as a clasp, strengthening the bond between Period and Cryptochrome.

These findings help to demonstrate how Period proteins act as a timekeeper that regulates how long Cryptochrome can turn down the activity of CLOCK and BMAL1. A deeper understanding of the molecular choreography among the four clock proteins holds promise for developing medications to treat the sleep disorders and circadian clock disruptions associated with a modern lifestyle.

the CLOCK-BMAL1-induced transcription of a luciferase reporter gene in cultured cells (*Griffin et al., 1999*; *Kume et al., 1999*). This transcriptional repression activity of CRYs likely occurs through their direct interactions with BMAL1 (*Griffin et al., 1999*; *Shearman et al., 2000*; *Partch et al., 2014*) and CLOCK (*Huang et al., 2012*). Despite the important repressor function of CRYs, the PER proteins have been suggested as the rate-limiting factor of the rhythmic negative feedback loop (*Lee et al., 2001*). With its protein abundance tightly regulated during the circadian cycle, PERs mediate the formation of the PER-CRY complexes and their nuclear localization. Once in the nucleus, PERs might physically bridge CRYs and CLOCK-BMAL1 and promote their interactions (*Chen et al., 2009*). The critical role of PERs in driving the molecular clock is underscored by the complete loss of circadian rhythmicity upon constitutive overexpression of PERs, but not CRY1, in vitro and in vivo (*Chen et al., 2009*; *McCarthy et al., 2009*; *Ye et al., 2011*).

Periodic degradation of PERs and CRYs represents another crucial step in the negative feedback loop. The F-box proteins, β-TrCP and FBXL3, have been discovered as the key ubiquitin ligases, responsible for promoting the polyubiquitination of PERs and CRYs, respectively (*Shirogane et al., 2005*; *Busino et al., 2007*; *Godinho et al., 2007*; *Reischl et al., 2007*; *Siepka et al., 2007*). Phosphorylation of a degron sequence serves as the signal for PER ubiquitination by β-TrCP (*Shirogane et al., 2005*), whereas recognition of CRYs by FBXL3 is made through a large protein-interaction interface without the involvement of a canonical degron motif or any post-translational modification (*Xing et al., 2013*). This CRY-FBXL3 interface is susceptible to disruption by both the CRY cofactor flavin adenine dinucleotide

(FAD) and the PER proteins, which have been suggested to control the stability of CRYs by directly competing with FBXL3 (*Xing et al., 2013*).

Although genetic studies have firmly established a central role of PERs in clock regulation, the molecular mechanisms by which PERs orchestrate the dynamic clock protein network remain elusive. Binding of PERs to CRYs, CLOCK, and BMAL1 have been detected both in vivo and in vitro (*Kiyohara et al., 2006*; *Ye et al., 2011*; *Partch et al., 2014*). However, the role of PER2 in coordinating the repression complex assembly is controversial. In addition, how PER–CRY interaction might interfere with FBXL3 for CRY binding also remains unclear. Here, we report the crystal structure of a PER2–CRY2 complex, which provides the missing structural framework for understanding the multiple functions of PERs in driving the molecular clock.

## Results

### Characterizing PER–CRY interactions

Mammalian PER1 and PER2 share ~50% sequence identity and a common domain architecture comprised of tandem N-terminal PER-ARNT-SIM (PAS) domains, a central CK1δ/ε-binding region, and a ~100 amino acid long C-terminal CRY-binding domain (CBD), which is necessary and sufficient for CRY binding (*Yagita et al., 2002*). The isolated PER2 CBD can stabilize CRY1/2 in vivo and compete with FBXL3 for CRY1/2 binding in vitro (*Chen et al., 2009*; *Xing et al., 2013*). In mouse embryonic fibroblasts (MEFs), overexpression of PER2-CBD alone was able to completely disrupt the circadian bioluminescence rhythm of the luciferase activity of a *Per^Luc* reporter gene (*Chen et al., 2009*). To first characterize the PER–CRY interaction, we performed an alanine-scanning mutagenic analysis of PER2-CBD. We initially targeted stretches of residues strictly conserved among vertebrate PER1/2 orthologs (*Figure 1A*, *Figure 1—figure supplement 1*). Surprisingly, none of the 10 single mutants, which were distributed along the length of the CBD, showed any detectable defect in CRY1 binding (*Figure 1—source data 1*). The PER2–CRY1 interaction was only abolished when alanine mutations were simultaneously introduced to two adjacent stretches of residues in the C-terminal, but not N-terminal half of PER2-CBD (*Figure 1B*). These results suggested an unusual binding mode of PER2-CBD onto CRYs and the importance of the C-terminal half of the CBD in complex formation.

### Overall structure

Mammalian CRY1 and CRY2 paralogs contain a highly similar photolyase-homology region (PHR) and a more diverse Cryptochrome C-terminal Extension (CCE) sequence (*Figure 1—figure supplement 1*). Their PER-binding activity has previously been mapped to the PHR, which is made of an α/β photolyase domain and an α-helical domain (*Figure 1C*). Consistent with their high sequence homology (86%), the crystal structures of CRY1-PHR and CRY2-PHR can be superimposed with a root-mean-square deviation (RMSD) of 0.43 Å out of 377 aligned Cα atoms. To gain structural insights into the general interaction between PERs and CRYs, we purified a representative PER2-CBD-CRY2-PHR complex and determined its crystal structure at a resolution of 2.8 Å (*Table 1*).

PER2-CBD adopts a highly extended structure, devoid of a hydrophobic core. It folds into five α-helices of variable length, which are dispersed along an otherwise linear polypeptide (*Figure 1C*). In the crystal, PER2-CBD meanders along one side of CRY2-PHR and sinuously wraps around the region. With nearly half of the PER2 residues involved in binding, the two proteins bury a total 2800 Å² of solvent accessible surface area at the interface, which stretches over a distance of more than 215 Å. This unusually extensive interface provides a plausible explanation for the high-affinity binding between the two clock proteins and their insensitivity to mutational disruption.

In comparison to its FBXL3-, KL001-, and FAD-complexed forms, CRY2 adopts the same global fold when bound to PER2-CBD (*Figure 4—figure supplement 2*). The largest structural variations take place in two local regions, the interface loop next to the FAD-binding pocket and a serine-rich loop neighboring a secondary pocket (see below). The majority of PER2-contacting residues on CRY2 (85%) are strictly conserved between mammalian CRY1 and CRY2, suggesting that the two cryptochrome proteins share a common PER2 binding mode.

### Interaction at the CRY2 C-terminal helix

The two stretches of residues, whose alanine mutations abrogated CRY1 binding, are mapped to a loop flanked by two α-helical regions in the C-terminal half of PER2 (*Figure 1D*). The PER2-CBD α3 helix preceding this loop packs against the long CRY2 C-terminal helix at an approximately 30° angle,

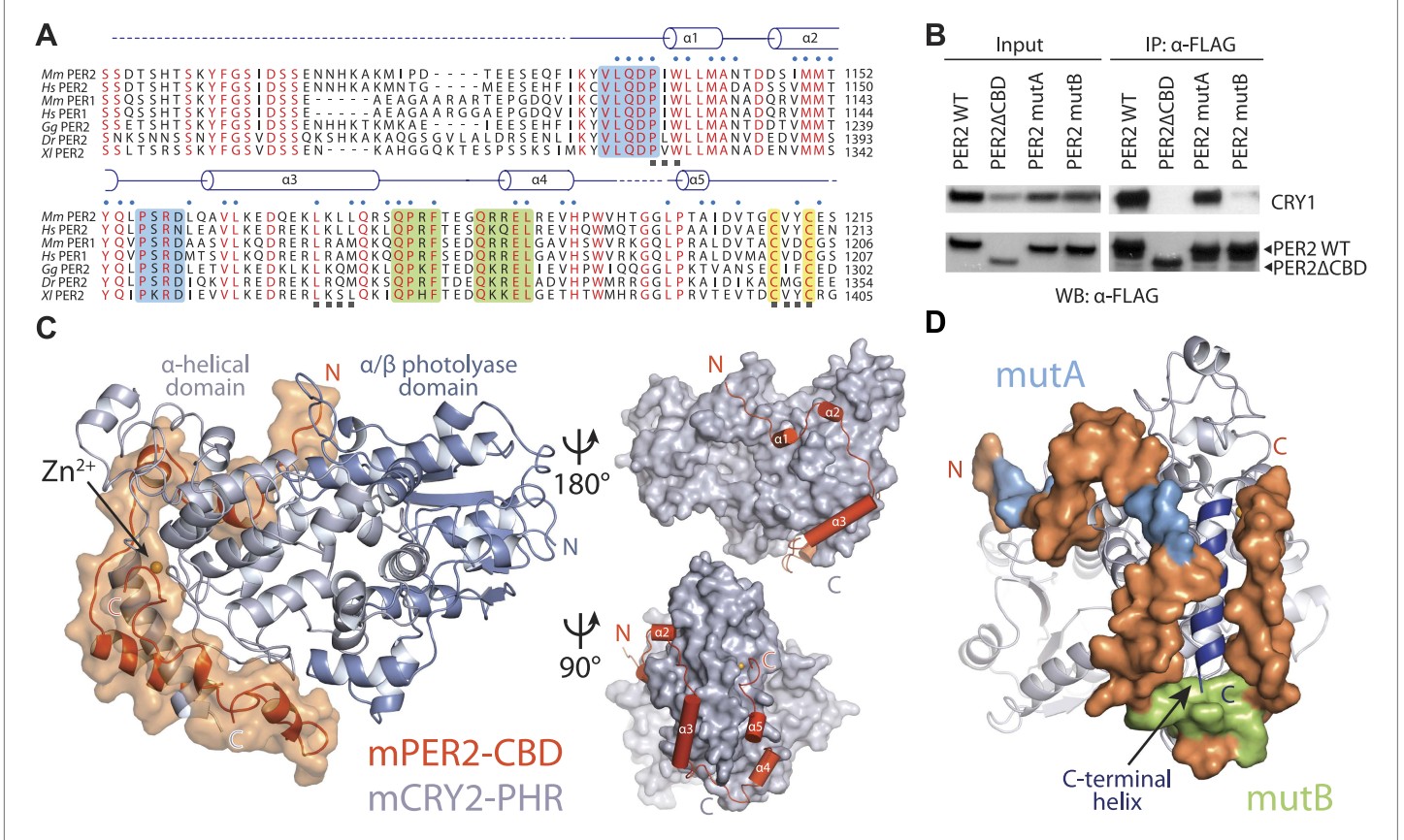

**Figure 1**. Overall structure of the murine CRY2–PER2 complex at 2.8 Å. (**A**) PER2 CBD sequence alignment. 49% of PER2 CBD residues interact with CRY2 (blue dots). The zinc-coordinating residues are conserved throughout vertebrates (highlighted in yellow). Blue and green boxes correspond to the mutA and mutB constructs, respectively, and indicate regions of PER2-CBD that were mutated to alanines. Dashed lines indicate crystallographically disordered regions. Black squares indicate residues mutated under structure guidance. (**B**) Co-immunoprecipitation of mutant PER2-CBD-FLAG constructs, only mutB was able to abolish CRY1-MYC binding. Western blot of an immunoprecipitaion of COS7 cells transfected with PER2-NLS-FLAG and CRY1-MYC. Proteins were precipitated with α-FLAG and then analyzed by Western blots using α-MYC and α-FLAG. (**C**) CRY2 PHR (gray) adopts an overall fold identical to its apo and complexed forms (e.g., FAD, FBXL3, and KL001). PER2 CRY-binding domain (CBD) (orange) shows a highly extended binding mode around CRY2. PER2 flanks the CRY2 C-terminal helix and coordinates a zinc ion with CRY2 within a CCCH-type intermolecular zinc finger motif. (**D**) Crystallographic data identify the location of alanine scanning mutants. Importantly, the mutB construct is centered around the CRY2 C-terminal helix.

The following source data and figure supplement is available for figure 1:

**Source data 1**. Alanine scanning mutants of PER2 CBD.

**Figure supplement 1**. Sequence alignment and structural elements of vertebrate CRY.

while the region C-terminal to the loop locks onto the same CRY2 helix from the other side (*Figure 1C–D*). Together, these PER2-CBD structural elements encircle the CRY2 C-terminal helix like an U-shaped clamp. Arg501 and Lys503 in the CRY2 C-terminal helix have previously been documented to be important for PER2 binding (*Ozber et al., 2010*). In the crystal, these two positively charged residues of CRY2 project in opposite directions and latch onto the surrounding PER2 regions by forming salt bridges with Asp1167 and Asp1206, respectively (*Figure 2A*). To confirm the critical role of the CRY2 C-terminal helix in binding PERs, we mutated two hydrophobic residues, Ile505 and Tyr506, at the end of this CRY2 helix, which are involved in fixing the α-helix to the rest of the CRY2 α-helical domain (*Figure 2B*). As expected, mutating both residues to aspartate completely abolished the PER2-binding activity of CRY2 (*Figure 2C*). The same effect was also achieved when negative charges were introduced to the side chains of a stretch of four nearby residues (amino acids 1171–1174) in the α3 helix of PER2-CBD (*Figure 2B*, *Figure 2—figure supplement 1A*). Based on these results, we conclude

**Table 1.** Data collection and refinement statistics

**CRY2-PER2**

| Data collection | |
|---|---|
| Space group | P41 |
| Cell dimensions | |
| *a, b, c (Å)* | 97.67, 97.67, 163.21 |
| α, β, γ (°) | 90, 90, 90 |
| Resolution (Å) | 2.9 (2.8) |
| $R_{meas}$ | 0.06 (0.8) |
| $I/\sigma I$ | 18.8 (2.1) |
| Completeness (%) | 99.6 (98.2) |
| Redundancy | 4.2 (4.2) |
| Refinement | |
| Resolution (Å) | 42.7–2.8 |
| No. reflections | 37541 (3671) |
| $R_{work}/R_{free}$ | 20.5/27.7 |
| No. atoms | 9342 |
| Protein | 9292 |
| Ligand/ion | 2 |
| Water | 48 |
| B-factors | 97.3 |
| Protein | 97.5 |
| Ligand/ion | 114.1 |
| Water | 66.3 |
| R.m.s. deviations | |
| Bond lengths (Å) | 0.009 |
| Bond angles (°) | 1.3 |

that the CRY2 C-terminal helix represents a key anchoring site for PER2 binding.

The close interaction between the C-terminal half of PER2-CBD and CRY2 C-terminal helix is immediately reminiscent of the docking mode between FBXL3 and CRY2. In the crystal structure of the FBXL3-CRY2 complex, the leucine-rich repeat (LRR) domain of FBXL3 engages CRY2 at the same site as PER2-CBD does in the PER2-CRY2 complex. The interface between FBXL3-LRR and CRY2 is also centered around the long C-terminal helix of the Cryptochrome protein. In fact, the CRY2 surface regions involved in contacting FBXL3-LRR and PER2-CBD share extensive overlapping regions (*Figure 2D*). Superposition analysis reveals that FBXL3 and PER2 cannot be simultaneously engaged with CRY2 without clashing into each other (*Figure 2—figure supplement 1B*). PERs, therefore, have the capability of protecting CRYs from FBXL3-mediated ubiquitination and degradation by directly competing with the ubiquitin ligase for binding CRYs.

## Intermolecular zinc finger

Amino acid sequence alignment of vertebrate PER1/2 orthologs reveals that their sequence conservation ends at a CXXC motif near the C-terminus (*Figure 1A*). In the complex structure, these two cysteine (C1210 and C1213) residues face a pair of cysteine and histidine residues in CRY2 (C432 and H491), which are also invariant among vertebrate CRY1/2 proteins (*Figure 1—figure supplement 1*). Together, these four residues sequester a strong density at the center, hinting at the coordination of a $Zn^{2+}$ ion at the end of the PER2–CRY2 interface (*Figure 3A*, *Figure 3—figure supplement 1A*). Indeed, we were able to validate the identity of the $Zn^{2+}$ ion by both anomalous dispersion measurements and inductively coupled plasma mass spectrometry (*Figure 3—figure supplement 1A,B*). Although a $Zn^{2+}$ ion has been previously reported to mediate protein–protein interactions (*Somers et al., 1994*), to our knowledge, this is the first CCCH-type intermolecular zinc finger that has been identified in a protein complex. Interestingly, the electron density of the PER2 sequence preceding the CXXC motif is not as strong as other regions of PER2-CBD, suggesting that the intermolecular zinc finger might have evolved to stabilize a flexible region of the PER–CRY interaction by acting as a 'molecular clasp'.

To assess the role of the intermolecular zinc finger in mediating PER–CRY association, we first tested the CRY2-binding activity of a recombinant mutant PER2-CBD, which lacks the CXXC motif. In comparison to the wild-type polypeptide, the ability of the PER2-CBD mutant to bind CRY2 was substantially compromised (*Figure 3B*). Similarly weakened interaction was also observed in a co-immunoprecipitation assay, in which the CXXC motif of the full-length PER2 protein or the two zinc-coordinating residues of CRY2 were mutated to alanines (*Figure 2—figure supplement 1A*, *Figure 3C*). Together, these results highlight the importance of the intermolecular zinc finger in strengthening the PER–CRY interface.

## Secondary pocket

Cryptochromes and DNA photolyases belong to the same superfamily of flavoproteins, whose common PHR fold is characterized by two large surface pockets, one for binding flavin adenine dinucleotide (FAD) and the other for binding a photoantenna cofactor, which is used by light-sensitive photolyases to catalyze FAD-dependent DNA repair (*Glas et al., 2009*; *Figure 4A*). Previously, we have identified

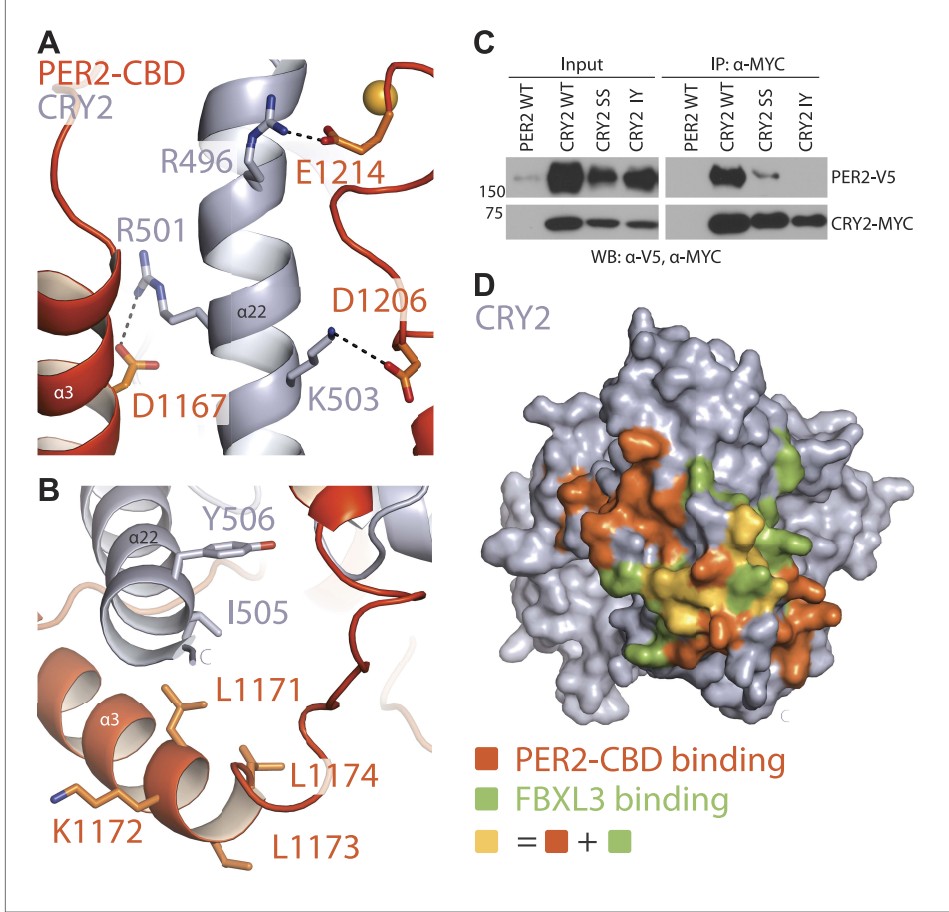

**Figure 2**. CRY2 C-terminal helix is the central locus of both PER2 and FBXL3 interactions. (**A**) PER2 (orange) forms three salt-bridges along CRY2 C-terminus helix (gray) R501 and K503 have been previously reported as critical binding residues. (**B**) A close-up view of the PER2-CRY2 interface at the end of CRY2 C-terminal helix. While the upper portion of the CRY2 C-terminal helix maintains ionic interactions with PER2, the lower is predominantly mediated by hydrophobic interactions. CRY2 and PER2 residues chose for subsequent mutational analysis are shown in sticks. (**C**) Concurrent mutations of hydrophobic residues on the CRY C-terminal helix (I505D and Y506D) prevent PER-CRY complex formation. Co-immunoprecipitations were performed with transfected full-length PER2-V5 and MYC-CRY2 in HEK293 cells with α-MYC beads and analyzed by Western blotting using α-V5 and α-MYC. See *Figure 2—figure supplement 1A* for corresponding PER2 mutants. (**D**) Surface mapping of FBXL3- and PER2-binding sites on CRY2. Residues that share contacts with PER2 and FBXL3 are colored in yellow and are clustered along the C-terminal helix. Other residues involved in binding PER2 and FBXL3 are colored in orange and green, respectively.

The following figure supplement is available for figure 2:

**Figure supplement 1**. Mutational and structural analysis of the PER2-CRY2 interface.

the FAD-binding pocket as a regulatory 'hot spot', which is targeted by FAD, the extreme carboxyl tail of FBXL3, and the clock-modulating small molecule, KL001 (*Nangle et al., 2013*; *Xing et al., 2013*; *Figure 4—figure supplement 1A*). However, the functional significance of the secondary pocket remained unexplored.

In the PER2-CRY2 crystal, the N-terminal half of PER2-CBD diverges from the FBXL3-binding site of CRY2 and reaches the rim of the secondary pocket after traversing around the α-helical domain (*Figure 4A*). With a highly conserved sequence, the N-terminal end of PER2-CBD is embedded in a V-shaped cleft formed between the two globular domains of CRY2-PHR, burying a PER2 tryptophan residue (Trp1139) at the junction (*Figure 4B*, *Figure 2—figure supplement 1A*). One side of the cleft is constructed by a serine-rich loop in CRY2, which we name as 'serine loop'. Distinct from its surrounding

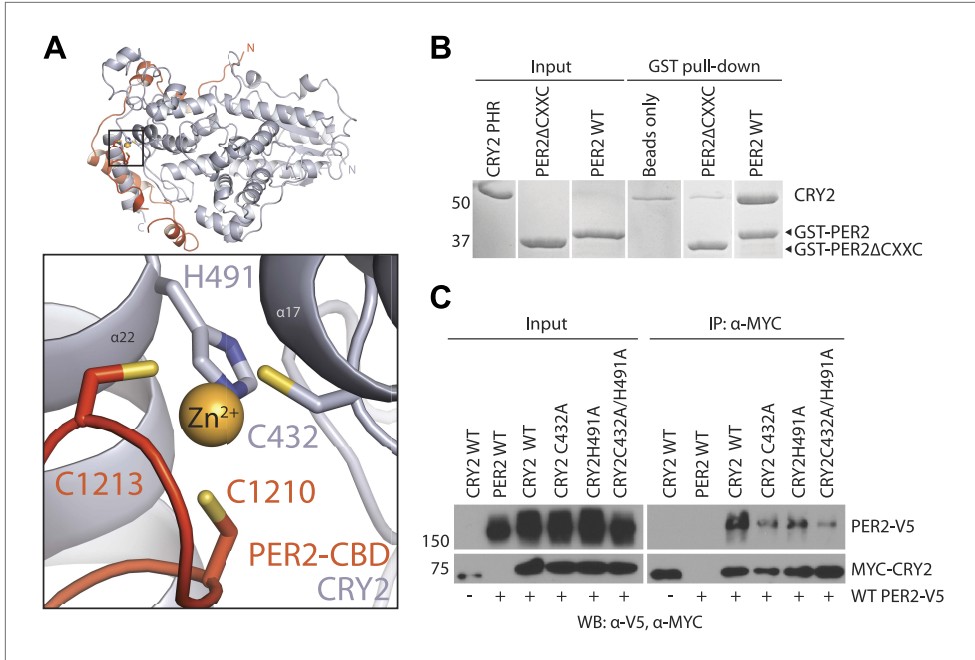

**Figure 3**. The intermolecular zinc finger is important for PER2–CRY2 complex formation. (**A**) Four conserved, contributing residues from PER2 (C1210 and C1213) and CRY2 (C432 and H491) form a CCCH-type zinc finger. (**B**) GST-pull-down assay with recombinant GST-tagged PER2ΔCXXC CBD and untagged CRY2-PHR protein show compromised CRY binding in the zinc finger mutant compared to WT PER2-CBD. (**C**) Similarly diminished interaction was replicated in a co-immunoprecipitation assay. Alanine mutations were introduced to CRY2 zinc-coordinating residues, C432 and H491, individually or in combination. Co-immunoprecipitations were performed with transfected full-length PER2 and CRY2 in HEK293 cells with α-MYC beads.

The following figure supplement is available for figure 3:

**Figure supplement 1**. Analysis of the intermolecular zinc finger.

regions, this loop adopts different conformations in several available crystal structures of CRY (*Figure 4—figure supplement 2*). Remarkably, PER2 binding induces yet another distinct structural configuration of the loop, thereby, defining a unique structural state of the local area next to the secondary pocket.

Although CRYs are known to not engage a second cofactor (*Zoltowski et al., 2011*; *Xing et al., 2013*), our previous cell-based random mutagenesis screen has identified three residues within this secondary pocket (Gly106 and Arg109 in CRY1, Glu121 in CRY2) (*Figure 4C*), whose missense mutations effectively abolished the repressor activity of CRYs (*McCarthy et al., 2009*). Among these three residues, Arg109 is exposed to the solvent and decorates one side of the pocket. Co-immunoprecipitation analysis of the R109Q mutant showed that alteration of this single amino acid is sufficient to abrogate CLOCK-BMAL1, but not PER1 or PER2 binding (*Figure 4D–F*). Thus, the secondary pocket of CRYs represents an important docking site for the heterodimeric transcriptional activators. Anchoring of PER2 at the edge of this CRY pocket not only reinforces its function as a previously unrecognized locus for protein–protein interactions, but also suggests a possible role of PERs in modulating the repressor functions of CRYs.

## Functional analysis of CRY mutants

To functionally characterize the multiple interfaces on CRYs mapped by the crystal structures, we systematically assessed several representative CRY mutants for their abilities to rescue rhythmicity in *Cry-deficient* MEF cells. Consistent with previous studies, wild-type CRY1 was able to repress the expression of the P(*Per2*)-*dLuc* reporter gene and produce robust bioluminescence rhythms. By contrast, the 'IY' mutant of CRY1, which confers severe structural disruption in the C-terminal helix, failed to restore any level of circadian rhythm, although it has the ability to repress CLOCK-BMAL1 as seen by the constitutively low luciferase signal (*Figure 5A*). Because the C-terminal helix of CRYs is a critical region for binding both FBXL3 and PERs, this result underscores the importance of CRY ubiquitination

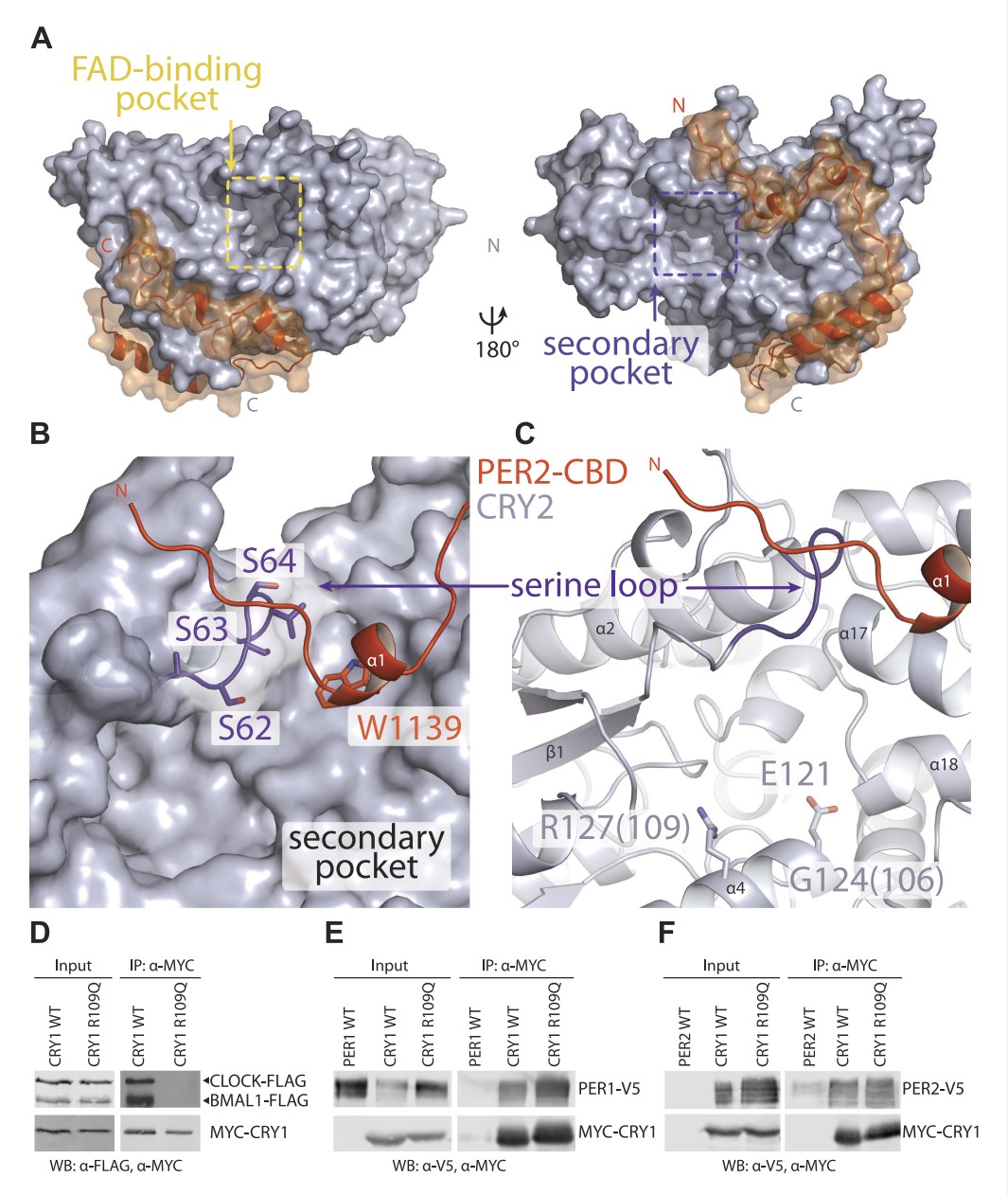

**Figure 4**. The secondary pocket is involved in CRY-CLOCK-BMAL1 complex assembly and repression. (**A**) Relative positions of the two large pockets on CRY2. (**B**) Surface representation of CRY2 with side chains of the serine loop shown in sticks. PER2 α1 helix inserts into a hydrophobic cleft. Compared to other CRY2 complexed forms, the serine loop flips up and engages PER2. (**C**) The serine loop lies opposite to the CRY α4 helix, which together frame the secondary pocket, the α4 helix contains three residues (CRY1 G106R and R109Q, CRY2 E121K), whose mutations result in a weak repression phenotype. (**D–F**) Co-immunoprecipitation assays show that the CRY1 R109Q mutant is unable to bind CLOCK-BMAL1, but retains PER1 and PER2 binding.

The following figure supplements are available for figure 4:

**Figure supplement 1**. Locations of structurally plastic loops on CRY2.

**Figure supplement 2**. CRY-PHR superposition: including CRY1 apo (red), CRY2 apo (light blue), KL001-bound (green), FAD-bound (orange), FBXL3-bound (cyan), and PER2-CBD-bound (gray) CRY.

**Figure supplement 3**. Major differences between CRY1-PER2-CBD and CRY2-PER2-CBD complex structures.

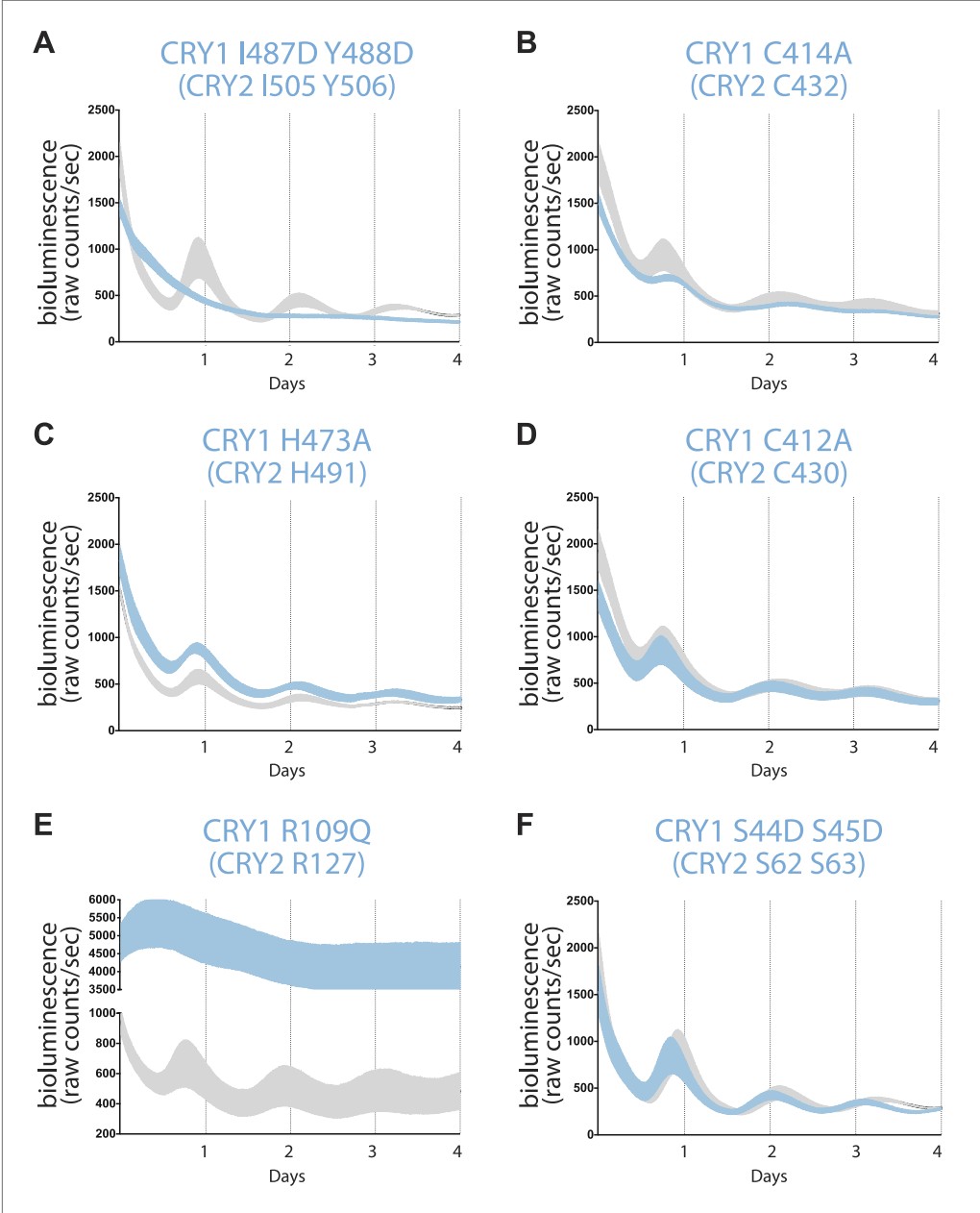

**Figure 5**. Real-time circadian rescue assays. *Cry1⁻ᐟ⁻/Cry2⁻ᐟ⁻* MEFs were transfected 24 hr after plating with *dLuc* reporter plasmid and *mCry1* expression or mutant vector. 72 hr after transfection, the cells were synchronized with dexamethasone. Bioluminescence (raw counts/s) monitoring was performed continuously for 70 s every 10 min using a photomultiplier tube at 37°C. Traces are shown as mean ± SEM and are representative of triplicate samples. Mutants are shown in blue and WT control in black. Only CRY1, not CRY2 is able to reconstitute robust circadian rhythmicity. (**A**) CRY1 I487D Y488D (CRY2 I505 Y506) 'IY' mutant abolishes rhythmicity but maintains repression compared to WT, suggesting that PER is not required for transcriptional repression. (**B** and **C**) Zinc-coordinating residues on CRY1 C414 and H473 (CRY2 C432 and H491) show blunted rhythm amplitude. (**D**) A nearby cysteine residue, C412 (CRY2 430), when mutated to alanine, does not show a significantly different phenotype from the WT control. (**E**) A critical residue on the secondary pocket, CRY1 R109 (CRY2 R127) shows a severely weakened repression phenotype when mutated to a glutamine. Traces are shown as mean ± SEM and are representative of duplicate samples. (**F**) Mutations of two serine residues in the serine loop, CRY1 S44D S45D (CRY2 S62 S63), show near WT rhythmicity and repression but with a 1-hr shorter period. For all mutants, corresponding CRY2 residues are in parenthesis.

and degradation in establishing clock rhythmicity and suggests the ability of CRY1 to inhibit CLOCK-BMAL1 in a PER-independent manner. In agreement, two CRY1 mutants unable to coordinate zinc, C414A and H473A, were also capable of transcriptional repression, even though their PER-binding activities are largely compromised (*Figure 5B,C*).

We noticed that the two zinc finger CRY1 mutants still sustained circadian rhythms. However, they showed defects in their bioluminescence oscillations (*Figure 5B,C*). Such a phenotype was not observed for a mutant with a nearby residue, Cys412, mutated to alanine, which did not perturb PER or FBXL3 binding as previously documented (*Figure 5D*; *Xing et al., 2013*). The contrast between the two zinc finger-defective mutants and the wild-type-like C412A mutant confirms the functional role of the zinc-coordinating residues in the negative arm of the feedback loop.

Consistent with its impaired CLOCK-BMAL1 binding activity, the CRY1 R109Q mutant showed significant derepression in the rescue assay (*Figure 5E*; *McCarthy et al., 2009*). This single amino acid mutation highlights the key role of the secondary pocket of CRYs for repression. Intriguingly, double serine to aspartate mutations (S44D S45D) in the nearby serine loop at the opposite side of the CRY secondary pocket completely rescued the circadian rhythm, although the period of the bioluminescence rhythms rescued by the mutant was reliably shorter than the wild-type CRY1 by about 1 hr (*Figure 5F*). In our co-immunoprecipitation experiments, this double serine mutation weakened PER2 binding to a lesser degree than the zinc finger mutations, which did not elicit a similar period-shortening effect (*Figure 2C*). Therefore, the period-shortening effect induced by the double serine mutation is likely specific to the defects of the local PER–CRY interface instead of their overall binding. It is conceivable that PERs might engage with CRYs near the CLOCK-BMAL1 docking site to control a periodicity-related step of negative feedback different from what they do at the predominant PER–CRY interface.

## Discussion

Previous studies have established a critical role of PERs in driving the rhythmic negative feedback loop (*Reppert and Weaver, 2002*). To fulfill this role, PERs have been suggested to act through multiple mechanisms, including mediating CRY nuclear entry, coupling CRYs to CLOCK-BMAL1, and competing with FBXL3 to stabilize CRYs. Our structural and mutagenic analyses of the PER2-CBD-CRY2 complex reveal a surprisingly robust binary assembly, which is resilient to mutational disruption. This stable complex is enabled by an extended binding mode of PER2-CBD, which spreads several distinct functional modules over a mostly linear interface. The hallmark of the PER–CRY interactions is its steric incompatibility with the FBXL3–CRY complex, which provides the structural basis for the competition of PERs and the FBXL3 ubiquitin ligase for controlling CRY stability. Interestingly, distant from the FBXL3–CRY interface, PERs also anchor themselves next to the putative CLOCK-BMAL1-binding pocket of CRYs, possibly regulating a specific step of transcriptional repression. Despite intensive genetic and cell-based studies, the precise spatial and temporal steps undertaken by PERs to coordinate transcriptional repression in the molecular clockwork remain unclear. On the one hand, PERs have been reported to be essential for CRYs to interact with CLOCK-BMAL1 (*Chen et al., 2009*). On the other hand, emerging evidence suggests that PERs binding might interfere with complex formation between CRYs and CLOCK-BMAL1 at certain steps during repression (*Ye et al., 2011*; *Akashi et al., 2014*). Conceivably, by interacting with the CRY C-terminal helix, PERs could compete with the C-terminus of the BMAL1 transactivation domain for CRY binding (*Czarna et al., 2011*). While detailed biochemical studies are necessary to resolve this controversy, our results offer the structural framework for in-depth mechanistic investigations.

Apart from the PER2-CBD-CRY2 complex, the crystal structures of CRY2 have been determined for four additional functional states, apo, FAD-, FBXL3-, and KL001-bound (*Nangle et al., 2013*; *Xing et al., 2013*). Together, these structures outline a rich landscape for the functional surfaces of mammalian CRYs, which distinguishes them from other members of the cryptochrome/photolyase superfamily. In their C-terminal α-helical domain, CRYs feature the conserved FAD-binding pocket, which is also targeted by the FBXL3 C-terminal tail and the clock-modulating drug, KL001. In their N-terminal α/β photolyase domain, CRYs have evolved the secondary pocket into a critical site for CLOCK-BMAL1 binding. Importantly, both CRY surface pockets are demarcated by structural elements with noticeable structural plasticity (*Figure 4—figure supplements 1 and 2*). The FAD-binding pocket is framed by the phosphate-binding loop and the interface loop on opposite edges, whereas the secondary pocket is guarded by the serine loop on one side. With the exception of the phosphate-binding loop, both the interface and serine loop have been shown to directly mediate protein–protein interactions.

Lastly, the extreme C-terminal α-helix of the mammalian CRYs presents yet another important surface area, which is responsible for the mutually exclusive binding of FBXL3 and PERs. Remarkably, all these molecular interacting sites likely represent an incomplete functional map of CRYs. Numerous mutants identified in our random mutagenesis screen of functionally deficient CRY1 and CRY2 bear mutations of amino acids located outside these sites (*McCarthy et al., 2009*). Future structural studies are needed to paint a complete picture of CRY functional surfaces.

Our crystal structure of the PER2-CBD-CRY2 complex unveils a structurally important intermolecular zinc finger, which might function as a stabilizing 'molecular clasp'. Although the evolutionary significance of the zinc-coordinating residues is apparent, as evidenced by their strict conservation across vertebrates, the functional significance of this unusual binding interface requires further investigation. On the one hand, the intermolecular zinc finger might be an intermediate product of the still evolving PER–CRY interface. On the other hand, it is plausible that this special protein interaction interface confers sensitivity to the fluctuating abundance of intracellular zinc (*Wang et al., 2012*), which might serve as a tissue-specific clock-modulating ion.

During the preparation of this manuscript, the complex structure of mammalian CRY1-PHR and PER2-CBD was reported (*Schmalen et al., 2014*). With high sequence conservation between CRY1 and CRY2, PER2-CBD adopts a similar CRY-binding mode with a tetrahedral coordination of a zinc ion by an intermolecular CCCH zinc-binding motif. The major structural difference lies at the interface of the N-terminal region of PER2-CBD and the CRY secondary pocket. The CRY1-bound PER2-CBD fragment contains a residual fusion-protein sequence, which forms an artifactual ß-hairpin with the first five amino acids of the PER2-CBD (*Figure 4—figure supplement 3*). In contrast to the PER2-bound CRY2 serine loop, but reminiscent of the *Drosophila* CRY antenna loop (*Zoltowski et al., 2011*), the otherwise disordered (*Czarna et al., 2013*) CRY1 serine loop adopts an inward conformation and occludes the secondary pocket. This conformational difference reveals a substantial degree of structural plasticity, which might be necessary for differential binding and regulation at this site. Interestingly, *Schmalen et al. (2014)* identified a potential redox sensor involving a disulfide bond near the zinc finger between Cys412 and Cys363, which modulates CRY1-PER2 binding. However, in our circadian reporter assay, we did not detect any difference between the CRY1 wild type and C412A mutant (*Figure 5D*). More in-depth analyses can now exploit the specific structural differences between the two complexes to explain the non-redundant roles of the two Cryptochrome proteins.

True to their name, Period proteins act as the master timekeepers in the circadian clock pathway, and likely use their multiple functional modules to simultaneously mediate the negative and positive phases of the clock through CRY stability and CRY-CLOCK-BMAL1 repression complex assembly.

## Materials and methods

### Recombinant protein purification

The mouse CRY2 (amino acids 1–512) was expressed as a glutathione S-transferase (GST) fusion protein in High Five (Invitrogen, Carlsbad, CA) suspension insect cells and isolated by glutathione affinity chromatography using buffer containing 20 mM Tris–HCl pH 8, 200 mM NaCl, 10% glycerol, 5 mM DTT (dithiothreitol). The protein was cleaved on-column by tobacco etch virus (TEV) protease then purified further by cation-exchange chromatography. Proteolytically stable murine PER2 (amino acids 1095–1215) was expressed as a GST-fusion protein in *Escherichia coli* expression system and isolated through glutathione affinity chromatography using buffer containing 20 mM Tris–HCl pH 8, 300 mM NaCl, 5 mM DTT. The protein was cleaved on-column by TEV protease then purified further by anion-exchange and size-exclusion chromatography. Both proteins were combined, concentrated, and further purified by size-exclusion chromatography using buffer containing 20 mM Tris–HCl pH 8, 300 mM NaCl, 5 mM DTT, 10% glycerol to establish stoichiometric binding.

### Crystallization, data collection, and structure determination

The crystals of the CRY2-PER2 complex were grown at 4°C by the hanging-drop vapor diffusion method, using 2 µl protein complex sample mixed 2:1 with reservoir solution containing 100 mM HEPES pH 7.5, 200 mM NaCl, 15% PEG 3350. Diffraction-quality crystals were subjected to a cryo-protectant procedure by gradually increasing the concentration of ethylene glycol to 25% (vol/vol) and then frozen in liquid nitrogen. The native and zinc anomalous data sets were collected at the BL8.2.1 beamline at the Advanced Light Source of the Lawrence Berkeley National Laboratory. Reflection data were

indexed, integrated, and scaled with the HKL2000 (*Otwinowski and Minor, 1997*). The CRY2-PER2 complex was determined by molecular replacement using CRY2 from the murine CRY2-KL001 complex structure (PDB:4MLP) as the search model. The structural models were manually built, refined, and rebuilt with the programs COOT (*Emsley et al., 2010*), PHENIX (*Adams et al., 2010*), and CCP4 (*Winn et al., 2011*). PER2 was built in following density modification. All figures were made using PyMOL (Schrödinger, LLC). Buried surface area was calculated using CNS (*Brunger et al., 1998*).

### In vitro GST pull down

GST-tagged mCRY2 (amino acids 1–512) was over-expressed in High Five insect cells suspension culture. GST-tagged mPER2 WT (amino acids 1095–1215) and GST-tagged mPER2ΔCXXC (amino acids 1095–1209) were over-expressed in *E. coli* and purified as previously described. Equal volumes CRY2-PHR was incubated with immobilized PER2 at 4°C for 1 hr. Glutathione beads were rigorously washed, and GST-PER2-CRY2 was released from the beads with SDS sample buffer, analyzed by SDS-PAGE and detected by Coomasssie stain.

### Co-immunoprecipitation

N-terminal Myc-tagged *Cry2* (0.25 µg) and a C-terminal V5-tagged *Per2* (0.5 µg) were transfected (Fugene 6, Madison, WI) into HEK293 cells. After 48 hr, cells were harvested and lysed by centrifugation. α-MYC-conjugated beads were used to immobilize MYC-CRY2. Beads were washed with buffer containing 50 mM Tris–HCl pH 7.5, 100 mM NaCl, 5% glycerol, 0.5 mM DTT, 0.5% Triton X-100, protease inhibitor (1:50). Protein was released from beads with SDS sample buffer and analyzed by Western blot using α-MYC and α-V5 for CRY2 and PER2, respectively.

### Real-time circadian rescue assays

Real-time circadian rescue assays performed as described in *Ukai-Tadenuma et al. (2011)*. $Cry1^{-/-}/Cry2^{-/-}$ MEFs were plated in 35-mm dishes at a density of $5 \times 10^5$ cells per dish. 24 hr later, cells were transfected with FuGene6 with 4 µg of pGL3-P(*Per2*)-dLuc reporter plasmid and 150 ng of the pMU2-*mCry1* expression vector (*Ukai-Tadenuma et al., 2011*) or mutant forms of this vector. 72 hr after transfection, the cells were synchronized by a 2-hr incubation in medium (DMEM/10% FBS/antibiotics) with dexamethasone (1 µM). The medium was then replaced with medium prepared from powdered DMEM without phenol red (Corning 90-013-PB) containing 4.5 g/l glucose and supplemented with 10 mM HEPES pH 7.2, 100 µM luciferin, 1 mM sodium pyruvate, 0.035% sodium bicarbonate, 10% FBS, antibiotics, and 2 mM L-glutamine. Bioluminescence monitoring was performed using a LumiCycle (Actimetrics, Inc. Wilmette, IL) to record from each dish continuously for ~70 s every 10 min using a photomultiplier tube at 37°C.

## Acknowledgements

We thank the beamline staff of the Advanced Light Source at the University of California at Berkeley for help with data collection. We also thank members of the Zheng laboratory and Wenqing Xu laboratory for discussion and help, and J Sean Yeung for ICP-MS analysis. We thank Drs Andrew Liu and Hiroki Ueda for their generous gift of the $Cry1^{-/-}/Cry2^{-/-}$ mouse embryonic fibroblasts and the *Cry1* rescue vector. This work is supported by the Howard Hughes Medical Institute (NZ and JST) and the National Institutes of Health (R01-CA107134 to NZ, GM090247 to CBG, and T32-GM007750 to SNN).

## Additional information

### Funding

| Funder | Grant reference number | Author |
| --- | --- | --- |
| Howard Hughes Medical Institute | | Joseph S Takahashi, Ning Zheng |
| National Institutes of Health | T32-GM007750 | Shannon N Nangle |
| National Institutes of Health | GM090247 | Carla B Green |
| National Institutes of Health | R01-CA107134 | Ning Zheng |

The funders had no role in study design, data collection and interpretation, or the decision to submit the work for publication.

## Author contributions

SNN, Conception and design, Acquisition of data, Analysis and interpretation of data, Drafting or revising the article; CR, NK, HT, Conception and design, Acquisition of data, Analysis and interpretation of data; JST, CBG, NZ, Conception and design, Analysis and interpretation of data, Drafting or revising the article

## Author ORCIDs

Joseph S Takahashi, http://orcid.org/0000-0003-0384-8878

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
