## [Decision Letter]

Thank you for sending your work entitled “Molecular Assembly of the Period-Cryptochrome Circadian Transcriptional Repressor Complex” for consideration at *eLife.* Your article has been favorably evaluated by a Senior editor, Louis Ptáček as Reviewing editor, and 2 reviewers, both of whom, Steve Kay and Aziz Sancar, have agreed to reveal their identity.

The Reviewing editor and the two reviewers discussed their comments before we reached this decision, and the Reviewing editor has assembled the following comments to help you prepare a revised submission.

The mammalian circadian clock is based on a transcription-translation feedback loop, in which CRY and PER proteins rhythmically inhibit the transcriptional activity of CLOCK:BMAL1. To date, the mechanistic roles of these two repressor proteins have remained largely unclear. Recently, several studies suggested that the molecular clock machinery relies on a sophisticated spatio-temporal complex assembly between CRY, PER and CLOCK:BMAL1. In this paper, the authors solved the crystal structure of mCRY2 (1-512) in complex with an mPER2 CRY-binding domain (CBD 1095-1215). Based on the structure information and mutagenesis analysis, the authors concluded that: (1) The N terminal of CBD is anchored at the edge of CRY secondary pocket that might be critical for the interactions with CLOCK:BMAL1, thus supporting the biochemical analysis of the clock protein interactions during the repressive phase of the TTFL. (2) Binding of PER2 to CRY interferes the interactions between FBXL3 and CRY. (3) A intermolecular Zinc finger motif stabilizes the CRY:PER complex. The paper is of good quality. The detailed interactions described between CRY and PER provide valuable information for future study to elucidate the molecular clock machinery.

Minor comments to address:

1) Figure 4—figure supplement 2 can be cited when the authors refer to the structural variations of the interface loop and a serine-rich loop.

2) Figure 2—figure supplement 1 can be cited when the authors refer to the role of PER2 Trp1139 (page 8, second paragraph) that may correspond to the PER2 PIW mutant.

3) In Figure 3, why is the band of CRY2H491A is missing in the input (lane 5, bottom panel)?

4) A crystal structure of mCRY1 (1-496) in complex with PER2 (1132-1252) has been published recently (Schmalen I et.al. 2014). The authors should cite this paper and comment on the similarities and differences between the two studies.

5) As mentioned by the authors, the interaction modes between PER, CRY and CLOCK:BMAL1 is controversial. The CRY1:PER2 structure paper from Eva Wolf lab (Schmalen I et.al. 2014) suggests that CRY C-terminal helix is involved in both the BMAL1 binding and PER binding supporting the improbability of a CRY-PER-CLOCK-BMAL1 complex. The authors may wish to comment on this point as well.

---

## [Author Response]

*1)*
Figure 4—figure supplement 2
*can be cited when the authors refer to the structural variations of the interface loop and a serine-rich loop*.

This citation has been added to the text.

*2)*
Figure 2—figure supplement 1
*can be cited when the authors refer to the role of PER2 Trp1139 (page 8, second paragraph) that may correspond to the PER2 PIW mutant*.

Yes, Trp1139 is the Trp in the PIW mutation. This citation has been added to the text.

*3) In*
Figure 3*, why the band of CRY2H491A is missing in the input (lane 5, bottom panel)*?

The input band is not visible due to a transfer error; as such we have repeated the experiment.

*4) A crystal structure of mCRY1 (1-496) in complex with PER2 (1132-1252) has been published recently (Schmalen I et.al. 2014). The authors should cite this paper and comment on the similarities and differences between the two studies*.

We have cited Schmalen I, et al. 2014 and added a brief comparison in the Discussion and in Figure 4—figure supplement 3 on the comparison between the two complexes. We have highlighted the overall similarity with the major differences seen at the CRY secondary pocket.

*5) As mentioned by the authors, the interaction modes between PER, CRY and CLOCK:BMAL1 is controversial. The CRY1:PER2 structure paper from Eva Wolf lab (Schmalen I et.al. 2014) suggests that CRY C-terminal helix is involved in both the BMAL1 binding and PER binding supporting the improbability of a CRY-PER-CLOCK-BMAL1 complex. The authors may wish to comment on this point as well*.

We have added a sentence and citation to [7] in the Discussion section commenting on the potential competition between PER and BMAL1 binding to the CRY C-terminal helix.